# Latent Template Induction with Gumbel-CRFs

**Yao Fu[1,*], Chuanqi Tan[2], Bin Bi[2], Mosha Chen[2], Yansong Feng[3], Alexander M. Rush[4]**

[1]ILCC, University of Edinburgh, [2]Alibaba Group, [3]WICT, Peking Univeristy, [4]Cornell University

yao.fu@ed.ac.uk, {chuanqi.tcq; b.bi; chenmosha.cms}@alibaba-inc.com

fengyansong@pku.edu.cn, arush@cornell.edu

## Abstract

Learning to control the structure of sentences is a challenging problem in text generation. Existing work either relies on simple deterministic approaches or RL-based hard structures. We explore the use of structured variational autoencoders to infer latent templates for sentence generation using a soft, continuous relaxation in order to utilize reparameterization for training. Specifically, we propose a Gumbel-CRF, a continuous relaxation of the CRF sampling algorithm using a relaxed Forward-Filtering Backward-Sampling (FFBS) approach. As a reparameterized gradient estimator, the Gumbel-CRF gives more stable gradients than score-function based estimators. As a structured inference network, we show that it learns interpretable templates during training, which allows us to control the decoder during testing. We demonstrate the effectiveness of our methods with experiments on data-to-text generation and unsupervised paraphrase generation.

## 1 Introduction

Recent work in NLP has focused on model *interpretability* and *controllability* [63, 34, 24, 55, 16], aiming to add transparency to black-box neural networks and control model outputs with task-specific constraints. For tasks such as data-to-text generation [50, 63] or paraphrasing [37, 16], interpretability and controllability are especially important as users are interested in what linguistic properties – e.g., syntax [4], phrases [63], main entities [49] and lexical choices [16] – are controlled by the model and which part of the model controls the corresponding outputs.

Most existing work in this area relies on non-probabilistic approaches or on complex Reinforcement Learning (RL)-based hard structures. Non-probabilistic approaches include using attention weights as sources of interpretability [26, 61], or building specialized network architectures like entity modeling [49] or copy mechanism [22]. These approaches take advantages of differentiability and end-to-end training, but does not incorporate the expressiveness and flexibility of probabilistic approaches [45, 6, 30]. On the other hand, approaches using probabilistic graphical models usually involve non-differentiable sampling [29, 65, 34]. Although these structures exhibit better interpretability and controllability [34], it is challenging to train them in an end-to-end fashion.

In this work, we aim to combine the advantages of relaxed training and graphical models, focusing on conditional random field (CRF) models. Previous work in this area primarily utilizes the score function estimator (aka. REINFORCE) [62, 52, 29, 32] to obtain Monte Carlo (MC) gradient estimation for simplistic categorical models [44, 43]. However, given the combinatorial search space, these approaches suffer from high variance [20] and are notoriously difficult to train [29]. Furthermore, in a linear-chain CRF setting, score function estimators can only provide gradients for the *whole* sequence, while it would be ideal if we can derive fine-grained pathwise gradients [44] for *each step* of the sequence. In light of this, naturally one would turn to reparameterized estimators with pathwise gradients which are known to be more stable with lower variance [30, 44].

Our simple approach for reparameterizing CRF inference is to directly relax the sampling process itself. Gumbel-Softmax [27, 38] has become a popular method for relaxing categorical sampling. We propose to utilize this method to relax each step of CRF sampling utilizing the forward-filtering backward-sampling algorithm [45]. Just as with Gumbel-Softmax, this approach becomes exact as temperature goes to zero, and provides a soft relaxation in other cases. We call this approach *Gumbel-CRF*. As is discussed by previous work that a structured latent variable may have a better inductive bias for capturing the discrete nature of sentences [28, 29, 16], we apply Gumbel-CRF as the inference model in a structured variational autoencoder for learning latent templates that control the sentence structures. Templates are defined as a sequence of states where each state controls the content (e.g., properties of the entities being discussed) of the word to be generated.

Experiments explore the properties and applications of the Gumbel-CRF approach. As a *reparameterized gradient estimator*, compared with score function based estimators, Gumbel-CRF not only gives lower-variance and fine-grained gradients for each sampling step, which leads to a better text modeling performance, but also introduce practical advantages with significantly fewer parameters to tune and faster convergence (§ 6.1). As a *structured inference network*, like other hard models trained with REINFORCE, Gumbel-CRF also induces interpretable and controllable templates for generation. We demonstrate the interpretability and controllability on unsupervised paraphrase generation and data-to-text generation (§ 6.2). Our code is available at `https://github.com/FranxYao/Gumbel-CRF`.

## 2 Related Work

**Latent Variable Models and Controllable Text Generation.** Broadly, our model follows the line of work on deep latent variable models [14, 30, 54, 23, 11] for text generation [28, 42, 16, 67]. At an intersection of graphical models and deep learning, these works aim to embed interpretability and controllability into neural networks with continuous [7, 67], discrete [27, 38], or structured latent variables [29]. One typical template model is the Hidden Semi-Markov Model (HSMM), proposed by Wiseman et al. [63]. They use a neural generative HSMM model for joint learning the latent and the sentence with exact inference. Li and Rush [34] further equip a Semi-Markov CRF with posterior regularization [18]. While they focus on regularizing the inference network, we focus on reparameterizing it. Other works about controllability include [55, 24, 33], but many of them stay at word-level [17] while we focus on structure-level controllability.

**Monte Carlo Gradient Estimation and Continuous Relaxation of Discrete Structures.** Within the range of MC gradient estimation [44], our Gumbel-CRF is closely related to reparameterization and continuous relaxation techniques for discrete structures [64, 36, 31]. To get the MC gradient for discrete structures, many previous works use score-function estimators [52, 43, 29, 65]. This family of estimators is generally hard to train, especially for a structured model [29], while reparameterized estimators [30, 54] like Gumbel-Softmax [27, 38] give a more stable gradient estimation. In terms of continuous relaxation, the closest work is the differentiable dynamic programming proposed by Mensch and Blondel [40]. However, their approach takes an optimization perspective, and it is not straightforward to combine it with probabilistic models. Compared with their work, our Gumbel-CRF is a specific differentiable DP tailored for FFBS with Gumbel-Softmax. In terms of reparameterization, the closest work is the Perturb-and-MAP Markov Random Field (PM-MRF), proposed by Papandreou and Yuille [46]. However, when used for sampling from CRFs, PM-MRF is a biased sampler, while FFBS is unbiased. We will use a continuously relaxed PM-MRF as our baseline, and compare the gradient structures in detail in the Appendix.

## 3 Gumbel-CRF: Relaxing the FFBS Algorithm

In this section, we discuss how to relax a CRF with Gumbel-Softmax to allow for reparameterization. In particular, we are interested in optimizing $\phi$ for an expectation under a parameterized CRF distribution, e.g.,

$$\mathbb{E}_{p_\phi(z|x)}[f(z)] \tag{1}$$

We start by reviewing Gumbel-Max [27], a technique for sampling from a categorical distribution. Let $Y$ be a categorical random variable with domain $\{1, .., K\}$ parameterized by the probability vector $\pi = [\pi_1, .., \pi_K]$, denoted as $y \sim \text{Cat}(\pi)$. Let $\text{G}(0)$ denotes the standard Gumbel distribution, $g_i \sim \text{G}(0), i \in \{1, .., K\}$ are i.i.d. gumbel noise. Gumbel-Max sampling of $Y$ can be performed as:

**Algorithm 1** Forward Filtering Backward Sampling

1: **Input:** $\Phi(z_{t-1}, z_t, x_t), t \in \{1, .., T\}, \alpha_{1:T}, Z$
2: Calculate $p(z_T|x) = \alpha_T / Z$
3: Sample $\hat{z}_T \sim p(z_T|x)$
4: **for** $t \leftarrow T - 1, 1$ **do**
5:    $p(z_t|\hat{z}_{t+1}, x) = \frac{\Phi(z_t, \hat{z}_{t+1}, x_{t+1})\alpha_t(z_t)}{\alpha_{t+1}(\hat{z}_{t+1})}$
6:    Sample $\hat{z}_t \sim p(z_t|\hat{z}_{t+1}, x)$
7: **end for**
8: **Return** $\hat{z}_{1:T}$

**Algorithm 2** Gumbel-CRF (Forward Filtering Backward Sampling with Gumbel-Softmax)

1: **Input:** $\Phi(z_{t-1}, z_t, x_t), t \in \{1, .., T\}, \alpha_{1:T}, Z$
2: Calculate:
3:    $\pi_T = \alpha_T / Z$
4:    $\tilde{z}_T = \text{softmax}((\log \pi_T + g)/\tau), g \sim \text{G}(0)$
5:    $\hat{z}_T = \text{argmax}(\tilde{z}_T)$
6: **for** $t \leftarrow T - 1, 1$ **do**
7:    $\pi_t = \frac{\Phi(z_t, \hat{z}_{t+1}, x_{t+1})\alpha_t(z_t)}{\alpha_{t+1}(\hat{z}_{t+1})}$
8:    $\tilde{z}_t = \text{softmax}((\log \pi_t + g)/\tau), g \sim \text{G}(0)$
9:    $\hat{z}_t = \text{argmax}(\tilde{z}_t)$
10: **end for**
11: **Return** $\hat{z}_{1:T}, \tilde{z}_{1:T}$    $\triangleright \tilde{z}$ is a relaxation for $\hat{z}$

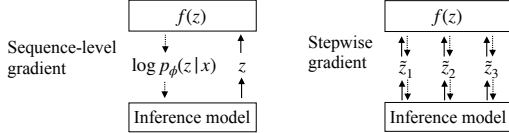

Figure 1: Gumbel-CRF FFBS algorithm and visualization for sequence-level v.s. stepwise gradients. Solid arrows show the forward pass and dashed arrows show the backward pass.

$y = \arg\max_i(\log \pi_i + g_i)$. Then the Gumbel-Softmax reparameterization is a continuous relaxation of Gumbel-Max by replacing the hard argmax operation with the softmax,

$$\tilde{y} = \text{softmax}([\log \pi_1 + g_1, .., \log \pi_K + g_K]) = \frac{\exp((\log \pi + g)/\tau)}{\sum_i \exp((\log \pi_i + g_i)/\tau)} \quad (2)$$

where $\tilde{y}$ can be viewed as a relaxed one-hot vector of $y$. As $\tau \to 0$, we have $\tilde{y} \to y$.

Now we turn our focus to CRFs which generalize softmax to combinatorial structures. Given a sequence of inputs $x = [x_1, .., x_T]$ a linear-chain CRF is parameterized by the factorized potential function $\Phi(z, x)$ and defines the probability of a state sequence $z = [z_1, .., z_T], z_t \in \{1, 2, ..., K\}$ over $x$.

$$p(z|x) = \frac{\Phi(z, x)}{Z} \qquad \Phi(z, x) = \prod_t \Phi(z_{t-1}, z_t, x_t) \quad (3)$$

$$\alpha_{1:T} = \text{Forward}(\Phi(z, x)) \qquad Z = \sum_i \alpha_T(i) \quad (4)$$

The conditional probability of $z$ is given by the Gibbs distribution with a factorized potential (equation 3). The partition function $Z$ can be calculated with the Forward algorithm [58] (equation 4) where $\alpha_t$ is the forward variable summarizing the potentials up to step $t$.

To sample from a linear-chain CRF, the standard approach is to use the forward-filtering backward-sampling (FFBS) algorithm (Algorithm 1). This algorithm takes $\alpha$ and $Z$ as inputs and samples $z$ with a backward pass utilizing the locally conditional independence. The hard operation comes from the backward sampling operation for $\hat{z}_t \sim p(z_t|\hat{z}_{t+1}, x)$ at each step (line 6). This is the operation that we focus on.

We observe that $p(z_t|\hat{z}_{t+1}, x)$ is a categorical distribution, which can be directly relaxed with Gumbel-Softmax. This leads to our derivation of the Gumbelized-FFBS algorithm(Algorithm 2). The backbone of Algorithm 2 is the same as the original FFBS except for two key modifications: (1) Gumbel-Max (line 8-9) recovers the *unbiased sample* $\hat{z}_t$ and the same sampling path as Algorithm 1; (2) the continuous relaxation of argmax with softmax (line 8) that gives the differentiable[2] (but biased) $\tilde{z}$.

We can apply this approach in any setting requiring structured sampling. For instance let $p_\phi(z|x)$ denote the sampled distribution and $f(z)$ be a downstream model. We achieve a reparameterized gradient estimator for the sampled model with $\tilde{z}$:

$$\nabla_\phi \mathbb{E}_{p_\phi(z|x)}[f(z)] \approx \mathbb{E}_{g \sim \text{G}(0)}[\nabla_\phi f(\tilde{z}(\phi, g))] \quad (5)$$

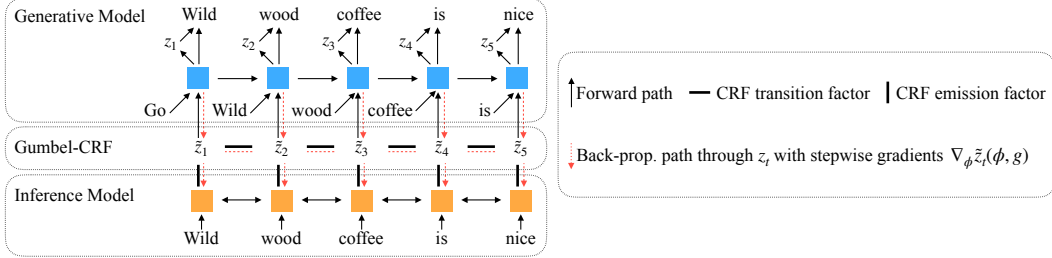

Figure 2: Architecture of our model. Note the structure of gradients induced by Gumbel-CRF differs significantly from score-function approaches. Score function receives a gradient for the sampled sequence $\nabla_\phi \log q_\phi(z|x)$ while Gumbel-CRF allows the model to backprop gradients along each sample step $\nabla_\phi \tilde{z}_t$ (red dashed arrows) without explicit factorization of the generative model.

We further consider a straight-through (ST) version [27] of this estimator where we use the hard sample $\hat{z}$ in the forward pass, and back-propagate through each of the soft $\tilde{z}_t$.

We highlight one additional advantage of this reparameterized estimator (and its ST version), compared with the score-function estimator. Gumbel-CRF uses $\tilde{z}$ which recieve direct *fine-grained gradients* for each step from the $f$ itself. As illustrated in Figure 1 (here $f$ is the generative model), $\tilde{z}_t$ is a differentiable function of the potential and the noise: $\tilde{z}_t = \tilde{z}_t(\phi, g)$. So during back-propagation, we can take *stepwise gradients* $\nabla_\phi \tilde{z}_t(\phi, g)$ weighted by the uses of the state. On the other hand, with a score-function estimator, we only observe the reward for the whole sequence, so the gradient is at the *sequence level* $\nabla_\phi \log p_\phi(z|x)$. The lack of intermediate reward, i.e., stepwise gradients, is an essential challenge in reinforcement learning [56, 66]. While we could derive a model specific factorization for the score function estimator [], this challenge is circumvented with the structure of Gumbel-CRF, thus significantly reducing modeling complexity in practice (detailed demonstrations in experiments).

# 4 Gumbel-CRF VAE

An appealing use of reparameterizable CRF models is to learn variational autoencoders (VAEs) with a structured inference network. Past work has shown that these models (trained with RL) [29, 34] can learn to induce useful latent structure while producing accurate models. We introduce a VAE for learning latent templates for text generation. This model uses a fully autoregressive generative model with latent control states. To train these control states, it uses a CRF variational posterior as an inference model. Gumbel CRF is used to reduce the variance of this procedure.

**Generative Model** We assume a simple generative process where each word $x_t$ of a sentence $x = [x_1, .., x_T]$ is controlled by a sequence of latent states $z = [z_1, .., z_T]$, i.e., template, similar to Li and Rush [34]:

$$p_\theta(x, z) = \prod_t p(x_t|z_t, z_{1:t-1}, x_{1:t-1}) \cdot p(z_t|z_{1:t-1}, x_{1:t-1}) \tag{6}$$

$$h_t = \text{Dec}([z_{t-1}; x_{t-1}], h_{t-1}) \tag{7}$$

$$p(z_t|z_{1:t-1}, x_{1:t-1}) = \text{softmax}(\text{FF}(h_t)) \tag{8}$$

$$p(x_t|z_t, z_{1:t-1}, x_{1:t-1}) = \text{softmax}(\text{FF}([e(z_t); h_t])) \tag{9}$$

Where $\text{Dec}(\cdot)$ denotes the decoder, $\text{FF}(\cdot)$ denotes a feed-forward network, $h_t$ denotes the decoder state, $[\cdot; \cdot]$ denotes vector concatenation. $e(\cdot)$ denotes the embedding function. Under this formulation, the generative model is autoregressive w.r.t. both $x$ and $z$. Intuitively, it generates the control states and words in turn, and the current word $x_t$ is primarily controlled by its corresponding $z_t$.

**Inference Model** Since the exact inference of the posterior $p_\theta(z|x)$ is intractable, we approximate it with a variational posterior $q_\phi(z|x)$ and optimize the following form of the ELBO objective:

$$\text{ELBO} = \mathbb{E}_{q_\phi(z|x)}[\log p_\theta(x, z)] - \mathcal{H}[q_\phi(z|x)] \tag{10}$$

Where $\mathcal{H}$ denotes the entropy. The key use of Gumbel-CRF is for reparameterizing the inference model $q_\phi(z|x)$ to learn control-state templates. Following past work [25], we parameterize $q_\phi(z|x)$

as a linear-chain CRF whose potential is predicted by a neural encoder:

$$h_{1:T}^{(\text{enc})} = \text{Enc}(x_{1:T}) \tag{11}$$

$$\Phi(z_t, x_t) = W_\Phi h_t^{(\text{enc})} + b_\Phi \tag{12}$$

$$\Phi(z_{t-1}, z_t, x_t) = \Phi(z_{t-1}, z_t) \cdot \Phi(z_t, x_t) \tag{13}$$

Where $\text{Enc}(\cdot)$ denotes the encoder and $h_t^{(\text{enc})}$ is the encoder state. With this formulation, the entropy term in equation 10 can be computed efficiently with dynamic programming, which is differentiable [39].

**Training and Testing**    The key challenge for training is to maximize the first term of the ELBO under the expectation of the inference model, i.e.

$$\mathbb{E}_{q_\phi(z|x)}[\log p_\theta(x, z)]$$

Here we use the Gumbel-CRF gradient estimator with relaxed samples $\tilde{z}$ from the Gumbelized FFBS (Algorithm 2) in both the forward and backward passes. For the ST version of Gumbel-CRF, we use the exact sample $\hat{z}$ in the forward pass and back-propogate gradients through the relaxed $\tilde{z}$. During testing, for evaluating paraphrasing and data-to-text, we use greedy decoding for both $z$ and $x$. For experiments controlling the structure of generated sentences, we sample a fixed MAP $z$ from the training set (i.e., the aggregated variational posterior), feed it to each decoder steps, and use it to control the generated $x$.

**Extension to Conditional Settings**    For conditional applications, such as paraphrasing and data-to-text, we make a conditional extension where the generative model is conditioned on a source data structure $s$, formulated as $p_\theta(x, z|s)$. Specifically, for paraphrase generation, $s = [s_1, ..., s_N]$ is the bag of words (a set, $N$ being the size of the BOW) of the source sentence, similar to Fu et al. [16]. We aim to generate a different sentence $x$ with the same meaning as the input sentence. In addition to being autoregressive on $x$ and $z$, the decoder also attend to [1] and copy [22] from $s$. For data-to-text, we denote the source data is formed as a table of key-value pairs: $s = [(k_1, v_1), ..., (k_N, v_N)]$, N being size of the table. We aim to generate a sentence $x$ that best describes the table. Again, we condition the generative model on $s$ by attending to and copying from it. Note our formulation would effectively become *a neural version of slot-filling*: for paraphrase generation we fill the BOW into the neural templates, and for data-to-text we fill the values into neural templates. We assume the inference model is independent from the source $s$ and keep it unchanged, i.e., $q_\phi(z|x, s) = q_\phi(z|x)$. The ELBO objective in this conditional setting is:

$$\text{ELBO} = \mathbb{E}_{q_\phi(z|x)}[\log p_\theta(x, z|s)] - \mathcal{H}[q_\phi(z|x)] \tag{14}$$

## 5   Experimental Setup

Our experiments are in two parts. First, we compare Gumbel-CRF to other common gradient estimators on the standard text modeling task. Then we integrate Gumbel-CRF to real-world models, specifically paraphrase generation and data-to-text generation.

**Datasets**    We focus on two datasets. For text modeling and data-to-text generation, we use the E2E dataset[50], a common dataset for learning structured templates for text [63, 34]. This dataset contains approximately 42K training, 4.6K validation and 4.6K testing sentences. The vocabulary size is 945. For paraphrase generation we follow the same setting as Fu et al. [16], and use the common MSCOCO dataset. This dataset has 94K training and 23K testing instances. The vocabulary size is 8K.

**Metrics**    For evaluating the gradient estimator performance, we follow the common practice and primarily compare the test negative log-likelihood (NLL) estimated with importance sampling. We also report relative metrics: ELBO, perplexity (PPL), and entropy of the inference network. Importantly, to make all estimates unbiased, all models are evaluated in *a discrete setting with unbiased hard samples*. For paraphrase task performance, we follow Liu et al. [37], Fu et al. [16] and use BLEU (bigram to 4-gram) [47] and ROUGE [35] (R1, R2 and RL) to measure the generation quality. We note that although being widely used, the two metrics do not penalize the similarity between the generated sentence and the input sentence (because we do not want the model to simply copy the input). So we adopt iBLUE [57], a specialized BLUE score that penalize the ngram overlap

Table 1: Density Estimation Results. NLL is estimated with 100 importance samples. Models are selected from 3 different random seeds based on validation NLL. All metrics are evaluated on the discrete (exact) model.

| Model | Neg. ELBO | NLL | PPL | Ent. | #sample |
|---|---|---|---|---|---|
| RNNLM | - | 34.69 | 4.94 | - | - |
| PM-MRF | 69.15 | 50.22 | 10.41 | 4.11 | 1 |
| PM-MRF-ST | 53.16 | 37.03 | 5.48 | 2.04 | 1 |
| REINFORCE-MS | 35.11 | 34.50 | 4.84 | 3.48 | 5 |
| REINFORCE-MS-C | 34.35 | 33.82 | 4.71 | 3.34 | 5 |
| Gumbel-CRF (ours) | 38.00 | 35.41 | 4.71 | 3.03 | 1 |
| Gumbel-CRF-ST (ours) | 34.18 | 33.13 | 4.54 | 3.26 | 1 |

| Estimators | Score /Reparam. | Seq. Level/ Stepwise | Unbiased MC Sample | Unbiased Grad. |
|---|---|---|---|---|
| REINFORCE-MS | Score | Seq. | Unbiased | Unbiased |
| REINFORCE-MS-C | Score | Seq. | Unbiased | Unbiased |
| PM-MRF | Reparam. | Step | Biased | Biased |
| PM-MRF-ST | Reparam. | Step | Biased | Biased |
| Gumbel-CRF | Reparam. | Step | Biased | Biased |
| Gumbel-CRF-ST | Reparam. | Step | Unbiased | Biased |

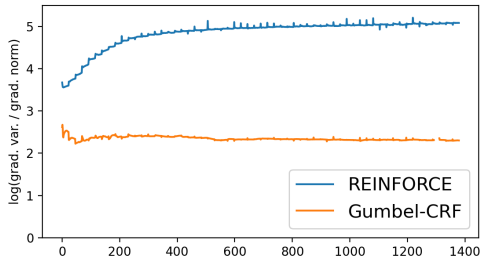

(A) Characteristics of the estimators we compare

(B) Log variance ratio, training curve

Figure 3: Text Modeling. (A). Characteristics of gradient estimators. (B). Variance comparison, Gumbel-CRF vs REINFORCE-MS, training curve.

between the generated sentence and the input sentence, and use is as our primary metrics. The iBLUE score is defined as: $\text{iB}(i, o, r) = \alpha \text{B}(o, r) + (1 - \alpha)\text{B}(o, i)$, where $\text{iB}(\cdot)$ denotes iBLUE score, $\text{B}(\cdot)$ denotes BLUE score, $i, o, r$ denote input, output, and reference sentences respectively. We follow Liu et al. [37] and set $\alpha = 0.9$. For text generation performance, we follow Li and Rush [34] and use the E2E official evaluation script, which measures BLEU, NIST [5], ROUGE, CIDEr [60], and METEOR [3]. More experimental details are in the Appendix.

**VAE Training Details**   At the beginning of training, to prevent the decoder from ignoring $z$, we apply word dropout [7], i.e., to randomly set the input word embedding at certain steps to be 0. After $z$ converges to a meaningful local optimal, we gradually decrease word dropout ratio to 0 and recover the full model. For optimization, we add a $\beta$ coefficient to the entropy term, as is in the $\beta$-VAE [23]. As is in many VAE works [7, 10], we observe the posterior collapse problem where $q(z|x)$ converges to meaningless local optimal. We observe two types of collapsed posterior in our case: a constant posterior ($q$ outputs a fixed $z$ no matter what $x$ is. This happens when $\beta$ is too weak), and a uniform posterior (when $\beta$ is too strong). To prevent posterior collapse, $\beta$ should be carefully tuned to achieve a balance.

## 6   Results

### 6.1   Gumbel-CRF as Gradient Estimator: Text Modeling

We compare our Gumbel-CRF (original and ST variant) with two sets of gradient estimators: score function based and reparameterized. For score function estimators, we compare our model with REINFORCE using the mean reward of other samples (MS) as baseline. We further find that adding a carefully tuned constant baseline helps with the scale of the gradient (REINFORCE MS-C). For reparameterized estimators, we use a tailored Perturb-and-Map Markov Random Field (PM-MRF) estimator [46] with the continuous relaxation introduced in Corro and Titov [9]. Compared to our Gumbel-CRF, PM-MRF adds Gumbel noise to local potentials, then runs a relaxed structured argmax algorithm [40]. We further consider a straight-through (ST) version of PM-MRF. The basics of these

Table 2: Data-to-text generation results. Upper: neural models, Lower: template-related models. Models are selected from 5 different random seeds based on validation BLEU.

| Model | BLEU | NIST | ROUGE | CIDEr | METEOR |
|---|---|---|---|---|---|
| D&J[13] | 65.93 | 8.59 | 68.50 | 2.23 | 44.83 |
| KV2Seq[15] | 74.72 | 9.30 | 70.69 | 2.23 | 46.15 |
| SUB[13] | 43.78 | 6.88 | 54.64 | 1.39 | 37.35 |
| HSMM[63] | 55.17 | 7.14 | 65.70 | 1.70 | 41.91 |
| HSMM-AR[63] | 59.80 | 7.56 | 65.01 | 1.95 | 38.75 |
| SM-CRF PC [34] | 67.12 | 8.52 | 68.70 | 2.24 | 45.40 |
| REINFORCE | 60.41 | 7.99 | 62.54 | 1.78 | 38.04 |
| Gumbel-CRF | 65.83 | 8.43 | 65.06 | 1.98 | 41.44 |

estimators can be characterized from four dimensions, as listed in Figure 3(A). The appendix provides a further theoretical comparison of gradient structures between these estimators.

Table 1 shows the main results comparing different gradient estimators on text modeling. Our Gumbel-CRF-ST outperforms other estimators in terms of NLL and PPL. With fewer samples required, reparameterization and continuous relaxation used in Gumbel-CRF are particularly effective for learning structured inference networks. We also see that PM-MRF estimators perform worse than other estimators. Due to the biased nature of the Perturb-and-MAP sampler, during optimization, PM-MRF is not optimizing the actual model. As a Monte Carlo sampler (forward pass, rather than a gradient estimator) Gumbel-CRF is less biased than PM-MRF. We further observe that both the ST version of Gumbel-CRF and PM-MRF perform better than the non-ST version. We posit that this is because of the consistency of using hard samples in both training and testing time (although non-ST has other advantages).

**Variance Analysis**     To show that reparameterized estimators have lower variance, we compare the log variance ratio of Gumbel-CRF and REINFORCE-MS-C (Figure 3 B), which is defined as $r = \log(\text{Var}(\nabla_\phi \mathcal{L})/|\nabla_\phi \mathcal{L}|)$ ($\nabla_\phi \mathcal{L}$ is gradients of the inference model)[3]. We see that Gumbel-CRF has a lower training curve of $r$ than REINFORCE-MS, showing that it is more stable for training.

## 6.2   Gumbel-CRF for Control: Data-to-Text and Paraphrase Generation

**Data-to-Text Generation**     Data-to-text generation models generate descriptions for tabular information. Classical approaches use rule-based templates with better interpretability, while recent approaches use neural models for better performance. Here we aim to study the interpretability and controllability of latent templates. We compare our model with neural and template-based models. Neural models include: D&J[13] (with basic seq2seq); and KV2Seq[15] (with SOTA neural memory architectures); Template models include: SUB[13] (with rule-based templates); hidden semi-markov model (HSMM)[63] (with neural templates); and another semi-markov CRF model (SM-CRF-PC)[34] (with neural templates and posterior regularization[18]).

Table 2 shows the results of data-to-text generation. As expected, neural models like KV2Seq with advanced architectures achieve the best performance. Template-related models all come with a certain level of performance costs for better controllability. Among the template-related models, SM-CRF PC performs best. However, it utilizes multiple weak supervision to achieve better template-data alignment, while our model is fully unsupervised for the template. Our model, either trained with REINFORCE or Gumbel-CRF, outperforms the baseline HSMM model. We further see that in this case, models trained with Gumbel-CRF gives better end performance than REINFORCE.

To see how the learned templates induce controllability, we conduct a qualitative study. To use the templates, after convergence, we collect and store all the MAP $z$ for the training sentences. During testing, given an input table, we retrieve a template $z$ and use this $z$ as the control state for the decoder. Figure 4 shows sentences generated from templates. We can see that sentences with different

| name: clowns \| eattype: coffee shop \| food: chinese \| customer_rating: 1 out of 5 \| area: riverside \| near: clare hall |
|---|

1. [there is a]$_{20}$ [coffee shop]$_{35}$ [in the]$_9$ [riverside]$_{35}$ [area ,]$_{12}$ [serves]$_{20}$ [chinese]$_{35}$ [food]$_{12}$ [. it is]$_{20}$ [called]$_{35}$ [clowns]$_{44}$ [. is]$_{20}$ [near]$_{35}$ [clare hall]$_{44}$ [. It has a customer rating]$_{20}$ [of 1 out of 5]$_8$ [.]$_{20}$

2. [clowns]$_{44}$ [is a]$_{20}$ [expensive]$_{12}$ [coffee shop]$_{35}$ [located]$_{12}$ [in]$_9$ [riverside]$_{35}$ [area]$_{12}$ [.]$_{20}$

3. [clowns]$_{44}$ [is a]$_{20}$ [coffee shop]$_{35}$ [in the riverside]$_9$ [. it is]$_{20}$ [family friendly]$_{12}$ [and has a]$_{20}$ [1]$_{45}$ [out of 5]$_8$ [stars]$_{12}$ [rating .]$_{20}$

| name: browns cambridge \| eattype: coffee shop \| food: chinese \| customer_rating: 1 out of 5 \| area: riverside \| familyfriendly: yes \| near: crowne plaza hote |
|---|

1. [browns cambridge]$_{44}$ [offers]$_{12}$ [chinese]$_{35}$ [food]$_{12}$ [near]$_{35}$ [crowne plaza hotel]$_{44}$ [in]$_{35}$ [riverside]$_{35}$ [. it is a]$_{20}$ [coffee shop]$_{35}$ [, not children friendly]$_{12}$ [and has a]$_{20}$ [5]$_{45}$ [out of 5]$_8$ [rating .]$_{20}$

2. [there is a]$_{20}$ [moderately priced restaurant]$_2$ [that serves]$_{20}$ [chinese]$_{35}$ [food]$_{12}$ [called]$_{35}$ [browns cambridge]$_{44}$ [coffee]$_9$ [. it has a customer rating]$_{20}$ [of 5 out of 5.]$_8$ [it is]$_{20}$ [not family-friendly]$_2$ [. it is]$_{20}$ [located]$_{12}$ [near]$_{35}$ [crowne plaza]$_{44}$

3. [browns cambridge]$_{44}$ [is a]$_{20}$ [chinese coffee shop]$_{35}$ [located]$_{12}$ [in]$_9$ [riverside near]$_{35}$ [crowne plaza hotel]$_{44}$ [. it has a]$_{20}$ [customer rating]$_{20}$ [of 5 out of]$_8$ [5]$_{44}$ [and is]$_{20}$ [not family-friendly]$_2$ [.]$_{20}$

Figure 4: Controllable generation with templates.

| Bigram | Sentence Segments | 4gram | Sentence Segments | |
|---|---|---|---|---|
| (A) 12-35 | 1. located near | (D) 35-44-12-20 | 1. near the city center | |
| | 2. restaurant near | | 2. near café rouge, there is a | |
| | 3. restaurant located near | | 3. in the city center, it is | ngrams w. semantically similar segments |
| (B) 20-8 | 1. has a customer rating of | (E) 44-20-35-20 | 1. french food at a moderate | |
| | 2. has a customer rating of 5 out of | | 2. french food for a moderate | |
| | 3. and with a customer rating of | | 3. fast food restaurant with a moderate | |
| (C) 20-12 | 1. is located | (F) 12-20-12-20 | 1. food with a price range of | ngrams w. semantically different segments |
| | 2. is a family friendly | | 2. price range and family friendly | |

Figure 5: Analysis of state ngrams. State ngrams correlate to sentence meaning. In cases (A, B, D, E), semantically similar sentence segments are clustered to the same state ngrams: (A) "location" (B) "rating" (D) "location" (E) "food" and "price". Yet there are also cases where state ngrams correspond to sentence segments with different meaning: (C1) "location" v.s. (C2) "comments"; (F1) "price" v.s. (F2) "price" and "comments".

templates exhibit different structures. E.g,. the first sentence for the *clowns* coffee shop starts with the location, while the second starts with the price. We also observe a *state-word correlation*. E.g,. state 44 always corresponds to the name of a restaurant and state 8 always corresponds to the rating.

To see how learned latent states encode sentence segments, we associate frequent $z$-state ngrams with their corresponding segments (Figure 5). Specifically, after the convergence of training, we: (a) collect the MAP templates for all training cases, (b) collapse consecutive states with the same class into one single state (e.g., a state sequence [1, 1, 2, 2, 3] would be collapsed to [1, 2, 3]), (c) gather the top 100 most frequent state ngrams and their top5 corresponding sentence segments, (d) pick the mutually most different segments (because the same state ngram may correspond to very similar sentence segments, and the same sentence segment may correspond to different state ngrams). Certain level of cherry picking happens in step (d). We see that state ngrams have a vague correlation with sentence meaning. In cases (A, B, D, E), a state ngram encode semantically similar segments (e.g., all segments in case A are about location, and all segments in case E are about food and price). But the same state ngram may not correspond to the same sentence meaning (cases C, F). For example, while (C1) and (C2) both correspond to state bigram 20-12, (C1) is about location but (C2) is about comments.

**Unsupervised Paraphrase Generation** Unsupervised paraphrase generation is defined as generating different sentences conveying the same meaning of an input sentence without parallel training instances. To show the effectiveness of Gumbel-CRF as a gradient estimator, we compare the results when our model is trained with REINFORCE. To show the overall performance of our structured model, we compare it with other unsupervised models, including: a Gaussian VAE for paraphrasing [7]; CGMH [41], a general-purpose MCMC method for controllable generation; UPSA [37], a strong paraphrasing model with simulated annealing. To better position our template model, we also report the supervised performance of a state-of-the-art latent bag of words model (LBOW) [16].

Table 3 shows the results. As expected, the supervised model LBOW performs better than all unsupervised models. Among unsupervised models, the best iB4 results come from our model trained with REINFORCE. In this task, when trained with Gumbel-CRF, our model performs worse than REINFORCE (though better than other paraphrasing models). We note that this inconsistency between the gradient estimation performance and the end task performance involve multiple gaps

Table 3: Paraphrase Generation. Upper: supervised models, Lower: unsupervised models. Models are selected from 5 random seeds based validation iB4 (iBLUE 4 gram).

| Model | iB4 | B2 | B3 | B4 | R1 | R2 | RL |
|---|---|---|---|---|---|---|---|
| LBOW [16] | - | 51.14 | 35.66 | 25.27 | 42.08 | 16.13 | 38.16 |
| Gaussian VAE[7] | 7.48 | 24.90 | 13.04 | 7.29 | 22.05 | 4.64 | 26.05 |
| CGMH [41] | 7.84 | - | - | 11.45 | 32.19 | 8.67 | - |
| UPSA [37] | 9.26 | - | - | 14.16 | 37.18 | 11.21 | - |
| REINFORCE | 11.20 | 41.29 | 26.54 | 17.10 | 32.57 | 10.20 | 34.97 |
| Gumbel-CRF | 10.20 | 38.98 | 24.65 | 15.75 | 31.10 | 9.24 | 33.60 |

Table 4: Practical benefits of using Gumbel-CRF. Typically, REINFORCE has a long list of parameters to tune: $h$ entropy regularization, $b_0$ constant baseline, $b$ baseline model, $r$ reward scaling, $\#s$ number of MC sample. Gumbel-CRF reduces the engineering complexity with significantly less parameters ($h$ entropy regularization, $\tau$ temperature annealing), less samples required (thus less memory consumption), and less time consumption. Models tested on Nvidia P100 with batch size 100.

| Model | Hyperparams. | #s | GPU mem | Sec. per batch |
|---|---|---|---|---|
| REINFORCE | $h, b_0, b, r, \#s$ | 5 | 1.8G | 1.42 |
| Gumbel-CRF | $h, \tau$ | 1 | 1.1G | 0.48 |

between ELBO, NLL, and BLEU. The relationship between these metrics may be an interesting future research direction.

**Practical Benefits**     Although our model can be trained on either REINFORCE or Gumbel-CRF, we emphasize that training structured variables with REINFORCE is notoriously difficult [34], and Gumbel-CRF substantially reduces the complexity. Table 4 demonstrates this empirically. Gumbel-CRF requires fewer hyperparameters to tune, fewer MC samples, less GPU memory, and faster training. These advantages would considerably benefit all practitioners with significantly less training time and resource consumption.

## 7   Conclusion

In this work, we propose a pathwise gradient estimator for sampling from CRFs which exhibits lower variance and more stable training than existing baselines. We apply this gradient estimator to the task of text modeling, where we use a structured inference network based on CRFs to learn latent templates. Just as REINFORCE, models trained with Gumbel-CRF can also learn meaningful latent templates that successfully encode lexical and structural information of sentences, thus inducing interpretability and controllability for text generation. Furthermore, the Gumbel-CRF gives significant practical benefits than REINFORCE, making it more applicable to real-world tasks.

## Broader Impact

Generally, this work is about controllable text generation. When applying this work to chatbots, one may get a better generation quality. This could potentially improve the accessibility [8, 53] for people who need a voice assistant to use an electronic device, e.g. people with visual, intellectual, and other disabilities [51, 2]. However, if not properly tuned, this model may generate improper sentences like fake information, putting the user at a disadvantage. Like many other text generation models, if trained with improper data (fake news [21], words of hatred [12]), a model could generate these sentences as well. In fact, one of the motivations for controllable generation is to avoid these situations [63, 12]. But still, researchers and engineers need to be more careful when facing these challenges.

## Footnotes

*Work done during an internship at Alibaba DAMO Academy, in collaboration with PKU and Cornell.

[2]Note that argmax is also differentiable almost everywhere, however its gradient is almost 0 everywhere and not well-defined at the jumping point [48]. Our relaxed $\tilde{z}$ does not have these problems.

[3]Previous works compare variance, rather than variance ratio [59, 19]. We think that simply comparing the variance is only reasonable when the scale of gradients are approximately the same, which may not hold in different estimators. In our experiments, we observe that the gradient scale of Gumbel-CRF is significantly smaller than REINFORCE, thus the variance ratio may be a better proxy for measuring stability.

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
