[Supplementary Material]

# Latent Template Induction with Gumbel-CRFs Appendix

## A  PM-MRF

As noted in the main paper, the baseline estimator PM-MRF also involve in-depth exploitation of the structure of models and gradients, thus being quite competitive. Here we give a detailed discussion.

Papandreou and Yuille [4] proposed the Perturb-and-MAP Random Field, an efficient sampling method for general Markov Random Field. Specifically, they propose to use the Gumbel noise to perturb each local potential $\Phi_i$ of an MRF, then run a MAP algorithm (if applicable) on the perturbed MRF to get a MAP $\hat{z}$. This MAP $\hat{z}$ from the perturbed $\tilde{\Phi}$ can be viewed as a biased sample from the original MRF. This method is much faster than the MCMC sampler when an efficient MAP algorithm exists. Applying to a CRF, this would mean adding noise to its potential at every step, then run Viterbi:

$$\tilde{\Phi}(z_t = i, x_{t-1}, x_t) = \Phi(z_t, x_{t-1}, x_t) + g, g \sim \text{G}(0) \quad \text{for all } t, i \tag{1a}$$

$$\hat{z} = \text{Viterbi}(\tilde{\Phi}) \tag{1b}$$

However, when tracing back along the Viterbi path, we still get $\hat{z}$ as a sequence of *index*. For continuous relaxation, we would like to relax $\hat{z}_t$ to be *relaxed one-hot*, instead of index. One natural choice is to use the Softmax function. The relaxed back-tracking algorithm is listed in Algorithm 1. In our experiments, for the PM-MRF estimator, we use $\tilde{z}$ for both forward and back-propagation. For the PM-MRF-ST estimator, we use $\hat{z}$ for the forward pass, and $\tilde{z}$ for the back-propagation pass.

It is easy to verify the PM-MRF is a biased sampler by checking the sample probability of the first step $\hat{z}_1$. With the PM-MRF, the biased $z_1$ is essentially from a categorical distribution parameterized by $\pi$ where:

$$\log \pi_i = \log \Phi(z_1 = i, x_1) \tag{2}$$

With forward-sampling, however, the unbiased $z_1$ should be from the marginal distribution where:

$$\log \pi_i = \log \beta_1(i) \neq \log \Phi(z_1 = i, x_1) \tag{3}$$

Where $\beta$ denote the backward variable from the backward algorithm [5]. The inequality in equation 3 shows that PM-MRF gives biased sample.

## B  Theoretical Comparison of Gradient Structures

We compare the detailed structure of gradients of each estimator. We denote $f(x_{1:n}, z_{1:n}) = \log p_\theta(x_{1:n}, z_{1:n})$. We use $\hat{z}$ to denote unbiased hard sample, $\tilde{z}$ to denote soft sample coupled with $\hat{z}$, $\hat{z}'$ to denote biased hard sample from the PM-MRF, $\tilde{z}'$ to denote soft sample coupled with $\hat{z}'$ output by the relaxed Viterbi algorithm. We use $w_{1:n}$ to denote the "emission" weights of the CRF. The gradients of all estimators are:

---

**Algorithm 1** Viterbi with Relaxed Back-tracking

---
1: **Input:** $\tilde{\Phi}(z_{t-1}, z_t, x_t), t \in \{1, .., T\}$
2: $s_1(i) = \log \tilde{\Phi}(i, x_1)$
3: **for** $t \leftarrow 2, T$ **do**
4:     $s_t(i) = \max_j\{s_{t-1}(j) + \log \tilde{\Phi}(z_{t-1} = j, z_t = i, x_t)\}$
5:     $b_t(i) = \text{Softmax}_j(s_{t-1}(j) + \log \tilde{\Phi}(z_{t-1} = j, z_t = i, x_t))$
6: **end for**
7: Back-tracking:
8: $\tilde{z}_T = \text{Softmax}(s_T)$
9: $\hat{z}_T = \text{Argmax}(s_T(i))$
10: **for** $t \leftarrow T - 1, 1$ **do**
11:     $\hat{z}_{t+1} = \text{Argmax}_i(\tilde{z}_{t+1}(i))$
12:     $\tilde{z}_t = b_{t+1}(\hat{z}_{t+1})$
13: **end for**
14: **Return** $\hat{z}, \tilde{z}$

---

$$\nabla_\phi \mathcal{L}_{\text{REINFORCE}} \approx \sum_t \underbrace{f(x_{1:n}, \hat{z}_{1:n})}_{\text{reward term}} \cdot \underbrace{\nabla_\phi \log q_\phi(\hat{z}_t | \hat{z}_{t-1}, x)}_{\text{stepwise term}} \tag{4}$$

$$\nabla_\phi \mathcal{L}_{\text{Gumbel-CRF-ST}} \approx \sum_t \underbrace{\nabla_{\tilde{z}_t} f(x_{1:n}, \hat{z}_{1:n})}_{\text{pathwise term}} \odot \underbrace{\nabla_\phi \tilde{z}_t(\hat{z}_{t+1}, w_{1:n}, \epsilon_t)}_{\text{stepwise term}} \tag{5}$$

$$\nabla_\phi \mathcal{L}_{\text{PM-MRF-ST}} \approx \sum_t \underbrace{\nabla_{\tilde{z}_t'} f(x_{1:n}, \hat{z}_{1:n}')}_{\text{pathwise term}} \odot \underbrace{\nabla_\phi \tilde{z}_t'(\hat{z}_{t+1}', w_{1:n}, \epsilon_t)}_{\text{stepwise term}} \tag{6}$$

In equation 4, we decompose $q(z|x)$ with its markovian property, leading to a summation over the chain where the same reward $f$ is distributed to all steps. Equations 5 and 6 use the chain rule to get the gradients. $\nabla_{\tilde{z}_t} f(x_{1:n}, \hat{z}_{1:n})$ denotes the gradient of $f$ evaluated on hard sample $\hat{z}_{1:n}$ and taken w.r.t. soft sample $\tilde{z}_t$. $\nabla_\phi \tilde{z}_t(\hat{z}_{t+1}, w_{1:n}, \epsilon_t)$ denotes the Jacobian matrix of $\tilde{z}_t$ (note $\tilde{z}_t$ is a vector) taken w.r.t. the parameter $\phi$ (note $\phi$ is also a vector, so taking gradients of $\tilde{z}_t$ w.r.t. $\phi$ gives a Jacobian matrix). Consequently $\odot$ is a special vector-matrix summation which result in a vector (note this is different with equation 4 since the later is a scalar-vector product). We further use $\tilde{z}_t(\hat{z}_{t+1}, w_{1:n}, \epsilon_t)$ to denote that $\tilde{z}_t$ is a function of the previous hard sample $\hat{z}_{t+1}$, all CRF weights $w_{1:n}$, and the local Gumbel noise $\epsilon_t$. Similar notation applies to equation 6.

All gradients are formed as a summation over the steps. Inside the summation is a scalar-vector product or a vector-matrix product. The REINFORCE estimator can be decomposed with a reward term and a "stepwise" term, where the stepwise term comes from the "transition" probability. The Gumbel-CRF and PM-MRF estimator can be decomposed with a pathwise term, where we take gradient of $f$ w.r.t. each sample step $\tilde{z}_t$ or $\tilde{z}_t'$, and a "stepwise" term where we take Jacobian w.r.t. $\phi$.

To compare the three estimators, we see that:
- **Using hard sample** $\hat{z}$. like REINFORCE, Gumbel-CRF-ST use hard sample $\hat{z}$ for the forward pass, as indicated by the term $f(x_{1:n}, z_{1:n})$
  - The advantage of using the hard sample is that one can use it to best explore the search space of the inference network, i.e. to search effective latent codes using Monte Carlo samples.
  - Gumbel-CRF-ST preserves the same advantage as REINFORCE, while PM-MRF-ST cannot fully search the space because its sample $\hat{z}'$ is biased.
- **Coupled sample path**. The soft sample $\tilde{z}_t$ of Gumbel-CRF-ST is based on the hard, exact sample path $\hat{z}_{t+1}$, as indicated by the term $\tilde{z}_t(\hat{z}_{t+1}, w_{1:n}, \epsilon_t)$.
  - The coupling of hard $\hat{z}$ and soft $\tilde{z}$ is ensured by our Gumbelized FFBS algorithm by applying gumbel noise $\epsilon_t$ to each transitional distribution $\tilde{z}_t = \text{Softmax}(\log q(z_t | \hat{z}_{t+1}, x) + \epsilon_t)$.
  - Consequently, we can recover the hard sample with the Argmax function $\hat{z}_t = \text{Argmax}(\tilde{z}_t)$.
  - This property allows us the use continuous relaxation to allow pathwise gradients $\nabla_\phi \tilde{z}_t(\hat{z}_{t+1}, w_{1:n}, \epsilon_t)$ without losing the advantage of using hard exact sample $\hat{z}$.

- PM-MRF with relaxed Viterbi also has this advantage of continuous relaxation, as shown by the term $\nabla_\phi \tilde{z}'_t(\hat{z}'_{t+1}, w_{1:n}, \epsilon_t)$, but it does not have the advantage of using unbiased sample since $\hat{z}'_{t+1}$ is biased.
- **"Fine-grained" gradients**. The stepwise term $\nabla_\phi \log q_\phi(\hat{z}_t | \hat{z}_{t-1}, x)$ in the REINFORCE estimator is scaled by the same reward term $f(x_{1:n}, \hat{z}_{1:n})$, while the stepwise term $\nabla_\phi \tilde{z}_t(\hat{z}_{t+1}, w_{1:n}, \epsilon_t)$ in the rest two estimators are summed with different pathwise terms $\nabla_{\tilde{z}_t} f(x_{1:n}, \hat{z}_{1:n})$.
  - To make REINFORCE achieve similar "fine-grained" gradients for each steps, the reward function (generative model) $f$ must exhibit certain structures that make it decomposable. This is not always possible, and one always need to manually derive such decomposition.
  - The fine-grained gradients of Gumbel-CRF is agnostic with the structure of the generative model. No matter what $f$ is, the gradients decompose automatically with AutoDiff libraries.

## C  Experiment Details

### C.1  Data Processing

For the E2E dataset, we follow similar processing pipeline as Wiseman et al. [6]. Specifically, given the key-value pairs and the sentences, we substitute each value token in the sentence with its corresponding key token. For the MSCOCO dataset, we follow similar processing pipeline as Fu et al. [1]. Since the official test set is not publically available, we use the same training/ validation/ test split as Fu et al. [1]. We are unable to find the implementation of Liu et al. [3], thus not sure their exact data processing pipeline, making our results of unsupervised paraphrase generation not strictly comparable with theirs. However, we have tested different split of the validation dataset, and the validation performance *does not change significantly with the split*. This indicates that although not strictly comparable, we can assume their testing set is just another random split, and their performance should not change much under our split.

### C.2  Model Architecture

For the inference model, we use a bi-directional LSTM to predict the CRF emission potentials. The dropout ratio is 0.2. The number of latent state of the CRF is 50. The decoder is a uni-directional LSTM model. We perform attention to the BOW, and also let the decoder to copy [2] from the BOW. For text modeling and data-to-text, we set the number of LSTM layers to 1 (both encoder and decoder), and the hidden state size to 300. This setting is comparable to [6]. For paraphrase generation, we set the number of LSTM layers (both encoder and decoder) to 2, and the hidden state size to 500. This setting is comparable to [1]. The embedding size for the words and the latent state is the same as the hidden state size in both two settings.

### C.3  Hyperparameters, Training and Evaluation Details

**Hyperparameters**  For the score function estimators, we conduct more than 40 different runs searching for the best hyperparameter and architecture, and choose the best model according to the validation performance. The hyperparameters we searched include: (a). number of MC sample (3, 5) (b). value of the constant baseline (0, 0.1, 1.0) (c). $\beta$ value ($5 \times 10^{-6}, 10^{-4}, 10^{-3}$) (d). scaling factor of the surrogate loss of the score function estimator (1, $10^2$, $10^4$). For the reparameterized estimators, we conduct more than 20 different runs for architecture and hyperparameter search. The hyperparameters we searched include: (a). the template in Softmax (1.0, 0.01) (b). $\beta$ value ($5 \times 10^{-6}, 10^{-4}, 10^{-3}$). Other parameter/ architecture we consider include: (a). number of latent states (10, 20, 25, 50) (b). use/ not use the copy mechanism (c). dropout ratio (d). different word drop schedule. Although we considered a large range of hyperparameters, we have not tested all combinations. For the settings we have tested, all settings are repeated 2 times to check the sensitivity under different random initialization. If we find a hyperparameter setting is sensitive to initialization, we run this setting 2 more times and choose the best.

**Training**  We find out the convergence of score-function estimators are generally less stable than the reparameterized estimators, they are: (a). more sensitive to random initialization (b). more prone to converging to a collapsed posterior. For the reparameterized estimators, the ST versions generally converge faster than the original versions.