[Reviews · NeurIPS 2020]

Review 1

Summary and Contributions: The paper proposes a continuous relaxation of CRF inference using the Gumbel softmax reparameterization, which makes it possible to backpropagate through the CRF if the CRF states are latent variables in a model. In an autoencoder setup the input is a bag of words (of the sentence to be generated), the state sequence act as a template for the output, and the sentence is generated as output. In neural variational inference for this model, the (neural) Gumbel-CRF is the inference network for the hidden states given the output and the expectation in the ELBO can be computed with dynamic programming. For text modelling, compared to strong reinforce-based models, Gumbel-CRF has lower variance and a higher ELBO. On data-to-text generation, performance is comparable to a previous Hidden semi-Markov template-based model, but lower than a recent template-based approach using posterior regularization and alignment heuristics for template induction. For unsupervised paraphrase generation, the model has lower Rouge but comparable BLEU to other techniques more customized for the task. Results are also similar to the Perturb and MAP CRF estimator on text modelling and paraphrase generation. --- Thanks for your response and clarifications.

Strengths: - The paper proposes a general-purpose approach for learning discrete latent variable models for sequence modeling that can be applied to controlled text generation. - The proposed approach is sound and is shown to give low-variance estimates as well as good performance on text generation tasks.

Weaknesses: - There are a number modelling details that are not entirely clear in the paper (see below). - When comparing Gumbel-CRF directly to Perturb-and-MAP MRF, performance is very similar.

Correctness: The claims and empirical methodology are correct.

Clarity: The paper is clearly written.

Relation to Prior Work: Previous work is discussed clearly, but a little more could be said on the relationship of the model with Perturb-and-MAP MRF.

Reproducibility: Yes

Additional Feedback: - For the experiments with the PM-MRF, is the same or a similar underlying model used to the Gumbel-CRF? - Can you give more details on how the straight-through estimator is used? - Based on the examples (and following previous work), for data-to-text generation it seems that a semi-Markov model is used (with variable length segments), but this isn't mentioned anywhere in the paper. - For the data-to-text generation, is there a model for predicting the templates from the (tabular) input, or where do the templates come from? Comparable previous work used a model for that, but it is not defined here.


Review 2

Summary and Contributions: In this paper, the authors proposed a reparametrization for sampling from linear-chain CRF and subsequent relaxation of such reparametrization that allows low-variance estimation of the corresponding gradient of the loss function with respect to parameters of the CRF. Specifically, the authors modified Forward-Filtering Backward-Sampling algorithm by replacing part of the algorithm that involves categorical distribution with Gumbel-softmax distribution. Using this estimator, the proposed Gumbel-CRF VAE model was used on a variety of text modelling tasks. The latent variable was represented by a so-called chain of templates, which is defined as a sequence of states where each state controls the content/properties of the word to be generated.

Strengths: Having the ability to develop interpretable and controllable text generation models is a crucial step towards developing transparent and fair NLP systems. The proposed relaxation of a linear-chain CRF is an exciting contribution and should be reasonably easy to implement using current automatic differentiation frameworks, hence it is likely going to be adopted by the community.

Weaknesses: I don't see any critical weaknesses in this work, however, I would like to point out some inaccuracies. 1. line 103, the authors write "the continuous relaxation of argmax with softmax gives the `differentiable` (but biased) sample". Although this is commonly mentioned in literature when explaining relaxations of the argmax, non-differentiability is not the reason for relaxing the argmax. In fact, the argmax is differentiable almost everywhere, but its discontinuities jump prevents swapping the order of expectation and differentiation operators. 2. The way the relaxation is described in algorithm 2 is still not a "complete" relaxation of FFBS procedure. The lines 5 and 9 (Alg.1) involve argmax function, and as previously discussed the gradient of this function is zero. In practice, this means that the gradient of \hat(z)_{t} with respect to \hat(z)_{t+1} is zero, which means that each sample receives an "immediate" gradient from the loss function defined on top of this samples, but doesn't receive "recurrent" part of the gradient. It seems that a proper relaxation would require relaxing forward variables and potentials (e.g. \alpha_{t+1}(z_{t+1}) in line 7 (alg.1 ) would be replaced with \sum_i \alpha_{t+1}(\hat{z}_{t+1}_i) * \hat{z}_{t+1}_i. Of course, later the straight-through estimator could be used, and essentially the algorithm would still use the hard sample, but in this case, "non-immediate"/"recurrent" gradients would not be zero. 3. Even though comparing models just using ELBO values is somewhat common, it is a bad practice, and it is not conclusive. The tightness of the bounds may be very different for different models. It would be helpful instead to check likelihood estimates using importance sampling.

Correctness: The claims and proposed method appear to be correct. However I would like the authors to address my remarks from the "Weaknesses" section.

Clarity: The paper is well written and easy to follow. Minor technical typo: there can't be an equality sign in eq.4, the relaxed gradient estimate is biased and can't be equal to the exact gradient. Also, in my opinion, the phrase "gradient estimator with low variance" if more suitable and accurate in comparison to "stable gradient estimator".

Relation to Prior Work: The authors clearly discussed how their contribution is related to previous work.

Reproducibility: Yes

Additional Feedback:


Review 3

Summary and Contributions: This paper presents a method for learning conditional random field (CRF) models with latent discrete variables using Gumbel-Softmax. The proposed method is then used to build a CRF-based variational autoencoder that allows one to learn a template for each sentence as a sequence of discrete latent values. Experimental results show that the Gumbel-Softmax approach gives better gradient estimates than existing methods such as REINFORCE. The CRF-based VAE is evaluated on a paraphrase generation task and a data-to-text generation task. ### The following is added after the author feedback. Most of my concerns have been resolved by the author feedback. I really hope that the authors incorporate those clarifications into the revised manuscript and do their best to improve the writing quality.

Strengths: The proposed Gumbel-Softmax-based approach is demonstrated to be superior to other approaches for learning CRFs with latent variables. The method can be implemented as a simple extension of the Forward Filtering Backward Sampling method and seems easy to implement.

Weaknesses: The experimental results on paraphrase generation and data-to-text generation are inconclusive. The results shown in Table 1 suggest that UPSA gives much better results than the proposed method. The results in Table 2 show that the proposed method is outperformed by an existing method. The authors claim that the proposed method has better interpretability and controllability but no systematic evaluation results about these two aspects are provided. The manuscript does not seem to have been proofread even by the authors and be ready for publication.

Correctness: The proposed approach seems technically sound.

Clarity: It is not clear how the experiments on paraphrase generation were conducted. How was the proposed model actually used to produce paraphrases? Was it given a bag of keywords or a sequence of discrete latent values for each sentence? What happens if the model ends up producing the same sentence as the input sentence? The experimental settings for the data-to-text task are not clear, either. What exactly is the input to the proposed model?

Relation to Prior Work: Relationships between the proposed method and relevant papers are discussed.

Reproducibility: No

Additional Feedback: Line 22: interpretability[27 -> itnerpretability [27 Line 23: modeling[50] -> modeling [50] Line 23: mechanism[23] -> mechanism [23] Line 40: backward-sampling -> backward-sampling (FFBS) Line 60: joint learning -> jointly leaning? Line 82: denotes -> denote Line 92: algorithm[58] -> algorithm [58] Line 97: operation that -> operation for which? Line 100: algorithm( -> algorithm ( Line 123: Where is M defined? Line 126: Perhaps the use of LSTMs should be mentioned somewhere around here. Line 131: approximation[7] -> approximation [7] Footnote: denotes the space -> denote the space Line 145: dropout[9] -> dropout [9] Line 146 and 149: local optimal -> local optimum? Line 147 and 148: as is in -> as in? Line 160: dataset[51] -> dataset [51] Line 174: we assume -> we assume that? Line 180 and 181: decoder to reconstruct -> decoder reconstruct Figure 3: number of sample -> number of samples? Figure 3: EBLO -> ELBO Line 192: this means -> this means that? Line 194: the the -> the Table 1: VAE[9] -> VAE [9] Table 2: D&J[15] -> D&J [15] Line 231: similar to -> comparably to? Line 237: sentences[17 -> sentences [17 Line 238: table to text -> table-to-text? Line 309: can i say -> can I say Line 348: lstm-crf -> LSTM-CRF Line 367: This paper has been published in ACL 2020. Line 384: Cgmh -> CGMH Line 388: monte carlo -> Monte Carlo Appendix Line 64: follow -> follow a Line 70: split -> splits Line 76: latent state -> latent states Line 77: to copy -> copy Line 110: are -> is


Review 4

Summary and Contributions: Neural template models [1, 2] are interesting with interpretability and controllability. Such models can be trained in a VAE framework with CRF as the posterior. Due to the non-differentiability of discrete templates, this paper investigates using Gumbel-Softmax as the gradient estimator for the posterior distribution against other gradient estimators such as REINFORCE and PM-MRF used by previous work. Empirically, the gumbel estimator demonstrates lower variance and better performance on unsupervised paraphrasing and data-to-text generation than comparable baselines. [1] Wiseman, Sam, Stuart M. Shieber, and Alexander M. Rush. "Learning Neural Templates for Text Generation." EMNLP 2018 [2] Xiang Lisa Li and Alexander M. Rush. Posterior control of blackbox generation. ACL 2020 ------------ After Rebuttal -------------- Thank you for the response! I think the response raises very nice points regarding the advantages of gumbel-crf over REINFORCE, such as less hyperparam tuning, less prone to posterior collapse, faster and stable training. However these points are not well-studied or discussed in the current form of the paper -- the current form only contains a bit discussion about variance in the toy non-autoregressive decoder setting. I am also concerned about the performance improvement present in the current form due to the use of post-processing decoder and access to the training data from the author response. I wonder why the model needs to access training instances in some way at test time, this makes the model non-parametric at test time and causes comparison fairness issues with other baselines. Overall, I suggest the authors can pay more focus on those mentioned advantages over REINFORCE in the next revision no matter this paper is accepted at NeurIPS or not. In my opinion, outperforming SOTA numbers is not very important for this paper, and it would be very strong if this paper can convincingly show the advantages over REINFORCE without losing performance. Given all the things mentioned about, I would like to stick to my original rating.

Strengths: 1. This paper studies an important and often difficult problem -- learning discrete structure from language without supervision. Properly learning such structures is usually hard due to non-differentiability and non-stable issues of gradient estimators. This paper adapts gumbel-softmax into this context and makes comparison with other estimators. 2. Experiments covers three unconditional/conditional text generation tasks. Quantitative results are also competitive.

Weaknesses: (1) I would say the paper lacks novelty on the method side -- REINFORCE and gumbel-softmax are very common gradient estimators when dealing with discrete variables. I think the major contributions are empirical results of application of gumbel-softax on this neural template model, but the experiment design has some flaws (see below) and the results are not very impressive to me. (2) The paper says it drops out all previous words to make the decoder depend on z on text modeling (line 171). Is this only for text modeling or for other tasks as well? And why is it necessary to do this? Dropping out all previous words seems a very big sacrifice to me and the necessity of doing that concerns me a bit -- I would expect the text modeling ability is much worse than an autoregressive LM, thus I won’t think this is a reasonable experiment setting if these non-autoregressive models are all bad baselines. I would like to see the log likelihood number from a regular LM in Figure 3(B) to see how large the gap is. (3) I wonder why a post-processing decoder is necessary here to refine the output since other baselines like HSMM seem not to use this. Is this because the model is non-autoregressive (which I am not sure)? This also raises my concern about comparison fairness when the proposed model uses post-processing decoder while other baselines did not (Table 2) (4) In paraphrase generation and data-to-text generation, an important comparison would be comparing the score estimator to show the advantage of gumbel-softmax against commonly used REINFORCE, which is the major claim in this paper. Currently the only evidence that supports gumbel-CRF performs better than score estimator is from the ELBO value in Figure 3(B), which is not very convincing to me as discussed in point (2) above.

Correctness: Yes they are correct.

Clarity: Yes this paper is very well-written

Relation to Prior Work: Yes this paper contains thorough discussion of related work

Reproducibility: Yes

Additional Feedback: Questions: How did the authors segment text as shown in Figure 4(B) given that x_t and z_t are one-to-one aligning? Additional feedback: Free-bits techniques might be worth trying for the posterior collapse issue which are studied in [1, 2] [1] Bohan Li et. al. A Surprisingly Effective Fix for Deep Latent Variable Modeling of Text. EMNLP 2019 [2] Tom Pelsmaeker and Wilker Aziz. Effective estimator of deep generative language models. ACL 2020.

[Author Response · NeurIPS 2020]

We thank all reviewers for their detailed constructive feedback and suggestions. Below are our responses:

**Clarify Technical Contributions (R3 / R4)**:

- **Gradient Estimation**: While the reviewers point out that REINFORCE and Gumbel-Softmax are now well-established techniques for discrete Categorical distributions (fixed $N$ sample space), when extending them to *structured / CRF* distributions (combinatorial complexity, $N^T$ sample space, $T$=sentence length), there are still many open challenges (e.g: seq-level. v.s. stepwise grad. Section 4, Appendix B.2; recurrent v.s. immediate grad. as pointed out by reviewer 2). We note that there is an active interdisciplinary effort on this task from multiple communities (NLP [1], Optimization [2], Probabilistic ML [3]). Our paper makes a targeted contribution about gradient structures in linear-chain CRFs for text generation and shows a novel use of Gumbel-Softmax for structured models.

- **Practical Benefits**: Training structured variables with REINFORCE is notoriously difficult [4], and Gumbel-CRF substantially reduces the complexity. Table B (below) demonstrates this empirically. Gumbel-CRF has: (a) fewer hyperparameters to tune; (b) less sensitivity to random seeds; (c) better gradient estimates; (d) less posterior collapse, especially for structured inference models (large amount of efforts in our experiments were to use multiple tricks to make REINFORCE work without posterior collapse). Table A (discussed below) further shows that these benefits persist in an auto-regressive setting. These advantages would considerably benefit all practitioners (just as Gumbel-Softmax has) with significantly less training time and resource consumption.

**Additional important concerns:**

- **NLL with importance sampling / autoregressive decoder (R2 / R4)**: We agree that using word dropout / non-autoregressive (NAR) decoder is not clearly motivated and orthogonal to the contribution of the paper. (This was originally done to compare with other latent-variable learning approaches.) Table A (below) gives the results for an autoregressive (AR) decoder for text modeling with NLL estimated by importance sampling. (ELBO results in the main paper Figure 3 did not add the constant term C in Equation 10, so we repeat the comparable NAR results here.) These experiments show that when trained with Gumbel-CRF, the AR decoder outperforms REINFORCE.

- **Clarification of terms and algorithms (R2)**: We apologize that using the term "differentiable z" may give the wrong implication as Argmax is differentiable almost everywhere [3]. We will clarify this in the paper. In terms of a full relaxation with the recurrent part, it could be implemented by changing line 7 in Algorithm 2 to an expectation weighted by $\tilde{z}_{t+1}$. Because we use a straight-through estimator, we want to recover an exact sample $\hat{z}$ with the Argmax in line 9 for the forward pass. In structured models, a fully relaxed sample path may diverge from the exact sample path. Our current relaxation couples the two. We will add more detailed discussions.

- **Modeling details (R1 / R3 / R4)**: For the text modeling experiments, we use the same underlying model for PM-MRF and Gumbel-CRF. For the straight-through estimator, we use the hard sample $\hat{z}$ in the forward pass, and the soft sample $\tilde{z}$ in the backward pass (source code torch_model_utils.py, line 293). For data-to-text generation, we use semi-Markov models as our baselines as had been done in previous work (Table 2). During testing, given a key-value pair, we find the training instance with the closest key under Jaccard distance, and use their templates for the given test case. For paraphrase generation, given a sentence, we retain its BOW, and retrieve a template from the training set with the closest BOW under Jaccard distance, and generate a new sentence. To get the segmentation in Figure 4, we collapse consecutive states into one state index, and report the state ngrams.

- **Comparison to SOTA models on paraphrasing and data-to-text (R3)**: Our method is orthogonal to many of the additional modeling techniques in SOTA models (e.g. posterior regularization in the SM-CRF model) so they can be integrated with ours. Although our model does not outperform existing SOTA models that use specifically designed techniques for each task, we aim to show that the approach scales and has auxiliary benefits (discussed above).

- **Comparing REINFORCE and Gumbel-CRF (R4)**: We unfortunately are not able to run new experiments comparing BLEU/ROUGE with REINFORCE. We believe with careful tuning, a REINFORCE model may perform well. However, even if the two perform similarly in metrics, the Gumbel-CRF shows significantly reduced modeling complexity (fewer hyperparameters and less sensitive to random seeds), making it a beneficial approach.

| | Dev | | Test | |
|---|---|---|---|---|
| | Neg. ELBO↓ | NLL↓ | Neg. ELBO↓ | NLL↓ |
| REINFORCE NAR | 73.85 | 73.82 | 73.81 | 73.79 |
| Gumbel-CRF NAR | 69.34 | 66.42 | 69.49 | 66.94 |
| RNNLM | – | 35.15 | – | 34.45 |
| REINFORCE AR Dec | 36.53 | 35.93 | 35.74 | 35.18 |
| Gumbel-CRF AR Dec | 36.85 | 34.95 | 35.87 | 34.19 |

Table A. NLL Comparison

| | Optimized Hyperaparameters | Dev NLL (6 random seeds under same hyper params.) | | | |
|---|---|---|---|---|---|
| | | mean | max | min | std |
| Gumbel-CRF NAR | Entropy regularization Softmax temperature | 68.28 | 66.42 | 69.23 | 1.01 |
| REINFORCE NAR | Entropy regularization Constant baseline Baseline model Reward scale Number of samples | 79.54 | 73.82 | 94.37 | 7.05 |

Table B. Hyperparameter Comparison and Sensitivity Analysis

[1] Hao et al. Backpropagating through Structured Argmax using a SPIGOT. ACL 2018

[2] Mensch and Blondel. Differentiable Dynamic Programming for Structured Prediction and Attention. ICML 2018

[3] Paulus et al. Gradient Estimation with Stochastic Softmax Tricks. Arxiv 2020

[4] Kim et al. Unsupervised Recurrent Neural Network Grammars. NAACL 2019


[Meta-Review · NeurIPS 2020]

This paper consider the text generation task in a VAE framework where the latent variables of a CRF are used as template of generation. The paper uses Gumbel-Softmax as the gradient estimator for the posterior distribution. As a reparameterized gradient estimator, the Gumbel-CRF gives more stable gradients than other gradient estimators such as REINFORCE and PM-MRF. The proposed method are tested in a variety of text modelling tasks. Reviewers agree this is an important and difficult problem. The paper gives a reasonable solution and experiments demonstrated its effectiveness. Although R3&R4 thought it is a bit lower than the throshold, their objection seems not very strong. I see the authors' response answered most of the questions. I would like to see the paper being accepted.